# Systemic Oncological Treatments versus Supportive Care for Patients with Advanced Hepatobiliary Cancers: An Overview of Systematic Reviews

**DOI:** 10.3390/cancers15030766

**Published:** 2023-01-26

**Authors:** Javier Bracchiglione, Gerardo Rodríguez-Grijalva, Carolina Requeijo, Marilina Santero, Josefina Salazar, Karla Salas-Gama, Adriana-Gabriela Meade, Alba Antequera, Ariadna Auladell-Rispau, María Jesús Quintana, Ivan Solà, Gerard Urrútia, Roberto Acosta-Dighero, Xavier Bonfill Cosp

**Affiliations:** 1Iberoamerican Cochrane Centre, Biomedical Research Institute Sant Pau (IIB Sant Pau), 08041 Barcelona, Spain; 2Interdisciplinary Centre for Health Studies (CIESAL), Universidad de Valparaíso, Viña del Mar 46383, Chile; 3CIBER Epidemiología y Salud Pública (CIBERESP), 28029 Madrid, Spain; 4Quality, Process and Innovation Direction, Valld’Hebron Hospital Universitari, Vall d’Hebron Barcelona Hospital Campus, 08035 Barcelona, Spain; 5Departament de Pediatria, d’Obstetrícia i Ginecologia, i Medicina Preventiva i Salut Pública, Universitat Autònoma de Barcelona, 08193 Barcelona, Spain

**Keywords:** liver neoplasms, biliary tract neoplasms, antineoplastic agents, molecular-targeted therapy, biological therapy, immunotherapy, review literature as topic

## Abstract

**Simple Summary:**

Hepatobiliary cancers (that include hepatocellular carcinoma, intrahepatic or extrahepatic cholangiocarcinoma and gallbladder cancer) are usually treated with systemic oncological treatments (i.e., chemotherapy, immunotherapy and biological or targeted therapies) mainly due to their improvement in survival. However, the trade-off between these therapies and usual practice supportive care is not clear, and other outcomes beyond survival should be considered in advanced stages, such as quality of life or symptom control. The present study is part of a wider project aiming to conduct broad evidence syntheses assessing the effects of systemic oncological treatments versus usual practice supportive care for patients with advanced non-intestinal digestive cancers. We performed an overview of systematic reviews assessing the effects of systemic oncological treatments versus usual practice supportive care for patients with primary advanced hepatobiliary cancer. We found evidence that for these patients (specifically for advanced hepatocellular carcinoma), systemic oncological treatments tend to improve survival at the expense of greater toxicity. Much of systematic reviews included was of low quality and highly overlapped. Nevertheless, the evidence we found failed to report other important outcomes that could be critical for decision making, including quality of life or symptom control. Future research assessing these patient-important outcomes is needed.

**Abstract:**

Background: The trade-off between systemic oncological treatments (SOTs) and UPSC in patients with primary advanced hepatobiliary cancers (HBCs) is not clear in terms of patient-centred outcomes beyond survival. This overview aims to assess the effectiveness of SOTs (chemotherapy, immunotherapy and targeted/biological therapies) versus UPSC in advanced HBCs. Methods: We searched for systematic reviews (SRs) in PubMed, EMBASE, the Cochrane Library, Epistemonikos and PROSPERO. Two authors assessed eligibility independently and performed data extraction. We estimated the quality of SRs and the overlap of primary studies, performed de novo meta-analyses and assessed the certainty of evidence for each outcome. Results: We included 18 SRs, most of which were of low quality and highly overlapped. For advanced hepatocellular carcinoma, SOTs showed better overall survival (HR = 0.62, 95% CI 0.55–0.77, high certainty for first-line therapy; HR = 0.85, 95% CI 0.79–0.92, moderate certainty for second-line therapy) with higher toxicity (RR = 1.18, 95% CI 0.87–1.60, very low certainty for first-line therapy; RR = 1.58, 95% CI 1.28–1.96, low certainty for second-line therapy). Survival was also better for SOTs in advanced gallbladder cancer. No outcomes beyond survival and toxicity could be meta-analysed. Conclusion: SOTs in advanced HBCs tend to improve survival at the expense of greater toxicity. Future research should inform other patient-important outcomes to guide clinical decision making.

## 1. Introduction

Primary hepatobiliary cancers (HBCs) are a group of neoplasms related to the liver, bile ducts or gallbladder, and include hepatocellular carcinoma (HCC), intrahepatic or extrahepatic cholangiocarcinoma and gallbladder cancer [1]. Hepatitis B virus, hepatitis C virus, alcohol abuse, non-alcoholic steatohepatitis and cirrhosis are known risk factors for these neoplasms with high variability among countries [2,3,4,5]. Currently, primary liver cancers (i.e., HCC and intrahepatic cholangiocarcinoma) are the sixth most commonly diagnosed cancers worldwide and were the third leading cause of cancer death in 2020, with an age-standardised incidence of 9.5 per 100,000 and a mortality of 8.7 per 100,000 [6,7], while gallbladder cancer has an age-standardised incidence and mortality of 1.2 and 0.8 per 100,000, respectively [6,7]. Based on current trends, it is estimated that liver cancers will have an overall increase of 58.6% in incidence and 60.9% in mortality worldwide by 2040, while gallbladder cancer will increase its incidence and mortality by 68.5% and 71.4%, respectively, in the same period [7]. Approximately 18.0% of liver cancers and 44.0% of gallbladder cancers are diagnosed in a distant stage [8,9]. Patients in these advanced stages have a poor overall prognosis, with a 1-year overall survival of 17.6% for liver cancers and 19.7% for gallbladder cancer and a 3-year overall survival of 5.0% and 3.7%, respectively [8,9].

Systemic oncological treatments (SOTs), such as chemotherapy, immunotherapy and targeted/biological therapies, constitute the widely used therapeutic approaches for patients with primary HBCs in advanced stages [10,11,12,13]. However, the reported recommendations are mainly based on survival-related outcomes and do not always consider (at least explicitly) all critical patient-centred outcomes, such as quality of life (QoL) or quality of end-of-life (EoL) care [14]. The EoL period has been conceptualised from a disease-centred perspective (‘a period of irreversible decline before death’) or from a time-based perspective (‘six months or less of life expectancy’) [15], but several authors have shifted these perspectives toward a person-centred one, considering the patients’ and their family’s perspectives when making decisions about treatments and overall care [16,17,18,19]. Considering this, the use of SOTs during the EoL period, among others, have been considered indicators of aggressive treatment and low-value medical practice [20,21,22,23].

Under this perspective, palliative care, usual practice supportive care (UPSC) or best supportive care (BSC), as defined in randomised controlled trials (RCTs), may be a valid therapeutic option for patients with advanced HBCs, not only as a complementary approach but also as a sole therapeutic alternative. Despite not having a consensual definition [24], UPSC can be broadly considered as all the efforts, treatments and techniques that aim to improve QoL and relieve symptoms [25,26]. It has repeatedly been shown to reduce suffering for patients and their families, and to lower public health costs [27,28,29].

Although SOTs have been reported to improve survival-related outcomes in patients with HBCs, whether this improvement is significant or clinically relevant in the EoL period, compared to UPSC, remains unclear. On the other hand, patients with advanced HBCs could benefit more from receiving palliative management aiming to relieve symptoms, increase their QoL and lower toxicity. In fact, every therapeutic decision should balance the increase in length of survival with maintaining a reasonable QoL, without being jeopardised by the burden of the treatment. In this sense, some authors have suggested a possible overuse of potentially inappropriate cancer care at the EoL, proposing quality indicators to provide healthcare policymakers with information to improve EoL care on a population level [30].

A broad evidence synthesis comparing the effects of SOTs and UPSC on survival-related outcomes, symptom-related outcomes, functional outcomes, toxicity, QoL and quality of EoL care could improve shared-decision-making processes between patients and practitioners in clinical practice. This study aims to identify, evaluate and summarise the evidence of all relevant systematic reviews (SRs) examining the benefits and harms of SOTs (chemotherapy, immunotherapy and targeted/biological therapies) versus UPSC in advanced HBCs.

## 2. Materials and Methods

The present study is part of a wider project aiming to conduct broad evidence syntheses assessing the effects of SOTs versus UPSC for patients with advanced non-intestinal digestive cancers [31,32,33]. This overview specifically addresses patients with advanced HBCs. We prospectively registered the protocol for this overview [31], adhering to the Preferred Reporting Items for Systematic Review and Meta-Analysis Protocols (PRISMA-P) guidelines [34]. The final version adheres to the Cochrane and the Preferred Reporting Items for Overviews of Reviews (PRIOR) guidelines [35,36].

### 2.1. Criteria for Considering Reviews for Inclusion

We used the PICOS framework (patients, intervention, comparison, outcomes, study type) to guide our eligibility criteria [37].

#### 2.1.1. Types of Studies

We included SRs assessing the clinical impact of SOTs on advanced HBCs, published from 2008 onwards (because one of the first landmark studies for our question was published this year). We considered an SR to be any type of secondary research published as full text that stated the following: (i) explicit eligibility criteria or research question; (ii) a structured search strategy (defined as explicit search terms and data frame, in at least two databases); (iii) explicit inclusion criteria and screening methods; (iv) an explicit assessment of the quality or risk of bias of each included study; and (v) an explicit approach to data analyses and synthesis. We considered eligible SRs conducting pairwise comparisons or network meta-analysis, including RCTs or observational studies. We excluded randomised clinical trials, quasi-experimental studies, observational studies, and descriptive studies. We also excluded clinical practice guidelines and reviews with no systematic methods according to our definition, such as narrative reviews.

#### 2.1.2. Types of Patients

We included SRs that considered adults (over 18 years of age) with any primary HBC, classified as advanced or metastatic by the authors of the study (stage IIIb, IIIc and IV for liver cancer, and IIIb and IV for bile duct cancer) at the time of receiving the treatment being evaluated. We excluded participants with lymphatic or stromal cancer, as well as hepatobiliary metastases from other locations.

#### 2.1.3. Types of Interventions and Comparators

As interventions, we considered any chemotherapy, immunotherapy or targeted/biological therapy, either given as monotherapy or in combination with other SOTs. We also allowed for inclusion interventions involving chemotherapy or other systemic treatments with concomitant palliative radiotherapy or prior surgery. We excluded reviews considering exclusively primary studies with only surgery or radiotherapy as an experimental intervention, or adjuvant or neoadjuvant chemotherapy therapies in the experimental arm.

We considered comparators to be any type of support treatment administered for symptomatic or palliative control, comprehending either usual treatment, UPSC or BSC [38]. Studies that did not specifically define the intervention of the control group were also included. Reviews that exclusively considered primary studies including some type of chemotherapy, biological/targeted therapy or immunotherapy in the control group were excluded. We also excluded comparisons with surgical or radiotherapeutic treatments with non-palliative intent. For reporting purposes, we generically referred to this group as ‘placebo/UPSC’.

#### 2.1.4. Type of Outcomes

As primary outcomes, we considered overall survival (OS) (measured as a dichotomous, continuous or time-to-event outcome), functional status (FS) (measured with Karnofsky or Eastern Cooperative Oncology Group [ECOG] scale) [39], quality of life (measured with any validated scale) and toxicity (measured as overall adverse events grade 3 or higher) [40]. Our secondary outcomes included progression-free survival (PFS), symptoms related to the disease (measured with any validated scale assessing one or more symptoms), admission to hospital or long-term centre or emergency consultations and quality of EoL care, defined as a composite outcome, including admission to the hospital at the EoL (in the last 30 days of life) and palliative care provided during the last year of life and place of death.

#### 2.1.5. Search Methods and Selection of Studies

We performed a first search strategy from inception until December 2019 in the following databases: Cochrane Database of Systematic Reviews; MEDLINE/PubMed; EMBASE/OVID; Epistemonikos; and PROSPERO. We did not use any language or publication status restrictions. Because this overview is part of a wider project, this initial search strategy included other types of non-intestinal digestive cancers [31]. Appendix A provides the detailed search strategy for MEDLINE/PubMed, which was adapted to the other databases. For the purposes of this overview, we updated the electronic search to August 2022, using a similar search strategy but considering only terms related to HBCs. Appendix A provides the detailed search strategy for this update. We also asked experts in the field for potentially relevant studies.

Two previously trained review authors (among JB, GRG, CR, MS, JS, KSG, AGM, AA, AAR, RAD, MJQ) performed an independent title and abstract screening of the results obtained from the search. A third review author (among JB, GRG, CR, MJQ) solved any disagreement. Afterwards, two review authors (among JB, GRG, CR, MS, JS, KSG, AGM, AA, AAR, RAD) conducted the full-text screening, with a third author (among JB, GRG, CR, MJQ) solving any disagreement. We used Covidence for the screening process www.covidence.org.

### 2.2. Data Extraction and Analysis

One review author (among JB, GRG, CR, MS, JS, KSG, AGM, AA, AAR, RAD) extracted data from the included SRs, using a previously piloted data extraction sheet. A second reviewer (among JB, GRG, CR, MS, JS, KSG, AGM, AA, AAR, RAD) cross-checked this process. We extracted both synthesised findings and disaggregated data for each included primary study concerning the outcomes of interest, as reported by the respective SR. We extracted data directly from the primary studies only if the SR did not clearly provide it.

One author (among JB, GRG) assessed the risk of bias (RoB) for each included SR using the AMSTAR-2 (A MeaSurement Tool to Assess systematic Reviews) tool [41]. A second author (among JB and GRG) cross-checked this assessment. We also described the RoB assessment of the primary studies made by the authors of each SR. We prioritised the reporting of assessments made using Cochrane RoB tools version 1. If two or more reviews conducted a contradictory assessment of the same primary study, we stated this was an unclear assessment.

We built a matrix of evidence for each type of included cancer to assess the possible overlap of primary studies within SRs. In this matrix, the columns represented all the included SRs, and the rows represented the primary studies included in each SR. We considered only the primary studies that provided useful data for our questions. Then, we calculated the overall corrected covered area (CCA), considering a CCA below 5% as slight overlap, >5% and <10% as moderate overlap, >10% and <15% as high overlap and >15% as very high overlap [42]. We also described a pairwise overlap assessment, that is, among each possible pair of SRs, considering the same thresholds and using the Graphical Representation of Overlap for OVErviews (GROOVE) tool, as described elsewhere [43].

We described the general characteristics and synthesised results of the included SRs. We also extracted the disaggregated data from each primary study, as reported by each SR. To avoid overestimating the effects by double counting the same study, if the same primary study was included in more than one review, we extracted the data only once. For each comparison, we performed a de novo meta-analysis based on the data of each primary study extracted from the SRs. We analysed dichotomous outcomes with risk ratio (RR), continuous outcomes with the mean difference or standardised mean difference and time-to-event outcomes with hazard ratios (HR). All of these had a 95% confidence interval (CI). Where available, we descriptively presented the results from the network meta-analysis to contrast the findings from pairwise comparisons.

We assessed the heterogeneity of the included studies with I^2^. We considered an I^2^ < 50% as low heterogeneity, >50% and <90% as high and >90% as very high [44]. If heterogeneity was below 90%, we performed a meta-analysis using a random-effects model. If heterogeneity was very high, we only described the results without performing a meta-analysis. We conducted all the analyses according to cancer localisation (liver versus bile duct) with a subgroup analysis according to the line of therapy (first versus second or more) in order to try to explain possible heterogeneity. We assessed the presence of possible publication bias by a visual inspection of a funnel plot for the primary outcomes if there were 10 or more included studies providing data for that specific comparison [45].

We assessed the certainty of the evidence according to GRADE guidance [46,47] for the following outcomes: (i) OS; (ii) symptoms related to the disease; (iii) FS; and (iv) QoL. We classified the certainty of the evidence for each outcome as high, moderate, low or very low. We explicitly stated if a specific outcome had no included studies. In that case, we did not assess the certainty of evidence. We also reported the main findings of the summary of findings (SoF) table in plain language, according to their specific assessment of the certainty of evidence.

## 3. Results

Our initial search strategy yielded a total of 2584 references. After removing duplicates, two reviewers assessed 2099 references by title and abstract, excluding 1894 references. Subsequently, two authors assessed 205 articles by full text, excluding 187 references, and finally including 18 SRs [48,49,50,51,52,53,54,55,56,57,58,59,60,61,62,63,64,65]. Appendix A provides the list of the excluded studies after full-text assessment, with their reasons. Figure 1 provides the PRISMA flow diagram summarising the screening process.

### 3.1. Description of Included Reviews

The included SRs were published between 2011 and 2022. Most of the SRs were conducted by research groups based in China [50,51,52,53,56,58,59], Canada [48,54,55,65] and the USA [49,61,62,64]. Among the included SRs, 11 (61.1%) performed pairwise meta-analyses [48,49,50,52,55,57,61,63,65] and seven (38.9%) conducted a network meta-analysis [51,56,58,59,62,64]. 

Table 1 provides an overall description of the included SRs. All the included SRs assessed only RCTs. Most studied patients with advanced HCC [49,50,51,52,53,54,55,56,57,59,60,61,62,63,64,65]. The most studied SOTs were biological/targeted therapies, especially sorafenib. The SRs considered SOTs mainly as a first- or second- line therapy. The authors did not provide an explicit definition for comparators, which included ‘placebo’, ‘UPSC’, ‘BSC’, ‘negative controls’ and ‘standard care’. Two SRs studied patients with advanced gallbladder cancer [48,58], but they included only one unique primary study assessing the effects of chemotherapy versus UPSC. None of the SRs assessed SOTs in patients with advanced cholangiocarcinoma.

### 3.2. Methodological Quality of the Included Reviews

In most cases, overall quality was classified as critically low [49,52,53,54,55,58,59,60,63,64,65] or low [51,56,57,61,62], mainly due to a lack of protocol registration and not reporting a list of the excluded studies. Table 2 provides a detailed assessment of each included review using the AMSTAR-2 tool. Appendix A provides a brief description of each unique primary study included within the reviews and the RoB assessment, as estimated by the SR authors.

### 3.3. Overlap Analysis

The included SRs comprehended a total of 22 unique trials relevant to our scope. Figure 2 presents an evidence matrix showing all the included SRs and the relevant primary studies for our overview. The SR that included most of the primary studies relevant for our scope was that conducted by Solimando et al. [60], while a total of nine SRs included only one [48,57,58,63] or two [49,52,53,56,65] primary studies relevant to our PICO question. All relevant primary studies were published between 2008 and 2020. Figure 3 provides the pairwise assessment of overlap among SRs using the GROOVE tool [44]. Overall overlap was very high for the whole matrix (CCA 22.2%, and 29.5% when adjusting by chronological structural missingness) [43,44]. The pairwise analysis of overlap revealed a total of 66 nodes with slight overlap, 10 nodes with moderate overlap, five nodes with high overlap and 72 nodes with very high overlap (including 11 pairs of reviews with 100% overlap). The SRs by Abdel-Rahman et al. and Jiang et al. were the only ones to assess gallbladder cancer [48,58]; therefore, it was expected that no overlap would occur between these reviews and the rest.

### 3.4. Effects of Interventions

The included SRs provided data for the following three comparisons: SOTs as first-line therapy versus placebo/UPSC in advanced HCC; SOTs as second-line therapy or more versus placebo/BSC in advanced HCC; and chemotherapy versus placebo/UPSC in advanced gallbladder cancer. Appendix A provides the SoF tables for the outcomes of interest within each comparison.

#### 3.4.1. Advanced Hepatocellular Carcinoma

##### SOTs as First-Line Therapy versus Placebo/UPSC

Eleven SRs [49,50,51,52,53,55,56,61,62,64,65], including four unique RCTs [66,67,68,69], assessed the effects of SOTs as first-line therapy versus placebo/UPSC. These trials assessed only biological/targeted therapies. Three trials included sorafenib as the intervention [66,67,68], and one trial assessed vandetanib as the intervention [69]. However, no data were available to be meta-analysed from this last study.

The overall pooled effect favoured the use of sorafenib over placebo/UPSC in terms of OS (HR = 0.62, 95% CI 0.55 to 0.77, high certainty). Sorafenib showed a higher rate of adverse events over placebo/UPSC (RR = 1.18, 95% CI 0.87 to 1.60, very low certainty). Figure 4 provides the forest plots for OS and toxicity for this comparison. There was not enough reported data to perform a meta-analysis for the other predefined outcomes.

None of the primary studies included within the SRs directly analysed a SOT other than sorafenib and vandetanib as a first-line treatment versus placebo/UPSC. However, the results of the indirect comparisons made in the network meta-analysis by Guo et al. [51] revealed that for patients without previous systemic treatment, lenvatinib and apatinib had the highest probability of achieving the best OS and PFS at one year, respectively. Sonbol et al. and Park et al. concluded that the combination of atezolizumab and bevacizumab was the best ranked treatment for both OS [62,64] and PFS as first-line therapy [64]. Sintilimab plus bevacizumab showed the best OS and PFS in the network meta-analysis by Liu et al. [56], with the combination of atezolizumab and bevacizumab ranking second.

##### SOTs as Second-Line Therapy or More versus Placebo/UPSC

Eleven SRs [50,51,54,55,57,59,60,61,62,63,64], including 17 unique RCTs [70,71,72,73,74,75,76,77,78,79,80,81,82,83,84,85,86], assessed the effects of SOTs as second-line therapy or more versus placebo/UPSC. SOTs studied by the primary studies included targeted/biological therapies (ADI-PEG20, apatinib, axitinib, brivanib, cabozantinib, everolimus, ramucirumab, regorafenib, sorafenib, tivantinib), immunotherapy (codrituzumab, pembrolizumab) and chemotherapy (S-1). Figure 5 shows the pooled effects of SOTs versus placebo/UPSC as second-line therapy or more for the outcomes OS, PFS and toxicity, disaggregated by subgroups according to the type of treatment. Overall, SOTs showed a beneficial pooled effect in terms of OS (HR = 0.85, 95% CI 0.79 to 0.92, moderate certainty) and PFS (HR = 0.67, 95% CI 0.55 to 0.80, low certainty). Toxicity was higher in the SOT group (RR = 1.58, 95% CI 1.28 to 1.96, low certainty). There was not enough reported data to perform a meta-analysis for the other predefined outcomes. One SR [54] narratively reported that QoL was similar for the comparison of regorafenib versus placebo/UPSC, and that the symptoms or participant functioning was similar when comparing ramucirumab versus placebo/UPSC.

Most network meta-analyses concluded that, as second-line therapy or more, regorafenib had the best OS ranking [51,60,64] and cabozantinib had the best PFS ranking as second-line treatments [60,62,64]. Guo et al. found that regorafenib also had the best PFS ranking [51]. Chen et al. suggests that regorafenib may have better OS in the subgroup of patients with low-level alpha-fetoprotein (<400 ng/mL) [59].

#### 3.4.2. Advanced Gallbladder Cancer

##### SOTs versus Placebo/UPSC for Advanced Gallbladder Cancer

Two SRs assessed the effects of chemotherapy versus placebo/UPSC [48,58], both including the same primary study conducted in patients with advanced gallbladder cancer [87]. This study compared two chemotherapy regimens (FUFA and Gemox) versus UPSC in 81 patients, showing no benefits of chemotherapy over UPSC in terms of OS at one year (RR = 0.89, 95% CI 0.74 to 1.07, very low certainty) [48]. Figure 6 provides the forest plot for OS at one year. No data were available for other outcomes or other types of SOTs.

## 4. Discussion

This overview identified, evaluated and summarised the evidence of 18 SRs examining the benefits and harms of SOTs versus UPSC in patients with advanced HBCs. Most of the SRs included patients with advanced HCC, were highly overlapped and had a low or critically low quality as assessed by the AMSTAR-2 tool. Our overview provided key findings for three main comparisons: (i) in patients with advanced HCC, sorafenib improves OS as first-line therapy compared to placebo/UPSC, but its effects on toxicity are still unclear; (ii) in patients with advanced HCC, compared to placebo/UPSC, SOT as a second-line therapy or more probably improves OS and may improve PFS, but with a higher toxicity; and (iii) in patients with advanced gallbladder cancer, compared to placebo/UPSC, it is uncertain if chemotherapy improves OS.

In our analysis, SOTs showed favourable effects in terms of survival compared with UPSC in patients with advanced HCC. As first-line therapy, three RCTs provided data on sorafenib versus placebo/UPSC, all showing better OS for the active treatment. Assuming a basal risk of death of 843 per 1000 patients at one year, treatment with sorafenib would reduce this to 683 per 1000. Because sorafenib was considered the standard of care for these patients, no research analysed other SOTs head-to-head against placebo/UPSC; therefore, all the evidence for this comparison comes from indirect comparisons. As a second-line therapy or more, cabozantinib, ramucirumab and regorafenib showed a statistically significant benefit in OS compared to placebo/UPSC. All other comparisons were non-significant, with a tendency to favour SOT, except for ADI-PEG20 and everolimus. These findings were consistent with those obtained by indirect comparison in network meta-analyses, which showed that regorafenib and cabozantinib may be the best options in this context. Certainty of evidence for adverse events within all comparisons was low or very low, but SOTs showed a tendency to have an overall higher toxicity than UPSC. However, some studies reported adverse events only for the experimental arm, which limited the analysis of these results.

Overall, our findings align with the recommendations made by the European Society for Medical Oncology (ESMO) and the American Society of Clinical Oncology (ASCO) guidelines [10,13] for patients with advanced HCC. ESMO recommends, for Barcelona Clinic Liver Cancer (BCLC) C patients, sorafenib as the standard of care as first-line therapy and regorafenib for those who have tolerated sorafenib but progressed, as well as UPSC for BCLC D patients. ASCO recommends atezolizumab-bevacizumab as first-line therapy (or sorafenib or lenvatinib in case of contraindications), and sorafenib, lenvatinib, cabozantinib, regorafenib or ramucirumab as second-line therapy. However, these recommendations do not include other patient-important outcomes, which is consistent with the identified gaps within the SRs. Only one of the included SRs explicitly stated quality of life and symptoms as outcomes, and none of the SRs assessed FS, hospital admissions or quality of EoL care as outcomes. This reporting of findings—centred around survival and toxicity—is consistent with classical recommendations made by expert panels about what endpoints to measure in clinical trials of HCC [88]. In this scenario where the risk–benefit balance is narrow, it is truly important to consider shared-decision-making processes and the values and preferences of patients. More recently, the inclusion of QoL and patient-reported outcomes has been recognised as an unmet need in clinical trials designed specifically for HCC [89], and core outcome sets for cancer, in general, have incorporated several outcomes beyond survival and toxicity, such as symptoms, QoL, overall health status, quality of care and economic outcomes [90,91]. A recent ESMO guideline also recommends monitoring patient-reported outcome measures for patients with cancer in an end-of-life context, focusing on symptom control and functional impairment [92]. Therefore, it is expected that future, high-quality RCTs and SRs will incorporate these outcomes [93,94] to ultimately enhance shared-decision-making processes.

### Strengths and Limitations

Overviews are inherently prone to miss recent primary studies not yet included in SRs. However, we consider that it is likely that we incorporated all bodies of evidence regarding HCC because, with the search update, we found recently published SRs. On the other hand, we found only two SRs for biliary duct cancers and none for cholangiocarcinoma, so it is possible we may have omitted relevant primary studies for these patients, or that there are no other primary studies conducted for this comparison; therefore, our results regarding this population should be interpreted carefully. The screening and data extraction processes were performed by two reviewers, but these stages may have been conducted by different pairs of reviewers. Because we conducted regular meetings throughout the conduction of this overview to standardise the criteria, and because all the reviewers were experienced, we think this issue should not have introduced an important bias in the selection of the studies and data extraction. Another limitation of our study was that the efficacy of each line of treatment could not be differentiated when two or more lines were administered. Although we conducted separate analyses by cancer location and type of treatment, some heterogeneity may persist in each intervention arm. There are different SOT schemes within each broad category of active treatments, and control arms include several different (usually poorly described) interventions. Nevertheless, we planned our methods to help clinicians and patients to have a broad overview of the available options, and not to necessarily provide information about specific treatments schemes. Furthermore, our study also has a meta-epidemiological role by identifying evidence gaps, highlighting and justifying the importance of conducting further research in certain areas (such as the generation of a systematic review regarding the effects of the addition of SOTs to UPSC for patients with advanced biliary tract cancer).

Otherwise, regarding the strengths in this overview, we thoroughly searched and selected relevant SRs from several databases, conducting a detailed analysis of their overall characteristics, quality and overlap. We also performed de novo meta-analyses for all relevant comparisons (avoiding the overestimation of the results due to double-counting data from primary studies), complementing the results with a description of the findings from network meta-analyses, and assessing the certainty of evidence according to GRADE guidelines.

## 5. Conclusions

Our findings show that in patients with advanced HCC, sorafenib is a valid therapeutic option as first-line therapy and other SOTs (cabozantinib, ramucirumab and regorafenib) as second-line therapy. However, these conclusions are mainly based on survival and toxicity outcomes. Future primary studies should consider and explicitly report other patient-important outcomes (such as QoL, symptom control and quality of EoL care), in order to provide useful data for evidence syntheses and clinical practice guidelines. Future SRs should also consider these outcomes in their protocols, planning to meta-analyse data from primary studies or explicitly report evidence gaps in primary research. Our overview has identified a gap in the evidence for comparisons of SOTs versus UPSC in patients with advanced bile duct cancers, for which conclusions regarding these patients are limited.

## Figures and Tables

**Figure 1 cancers-15-00766-f001:**
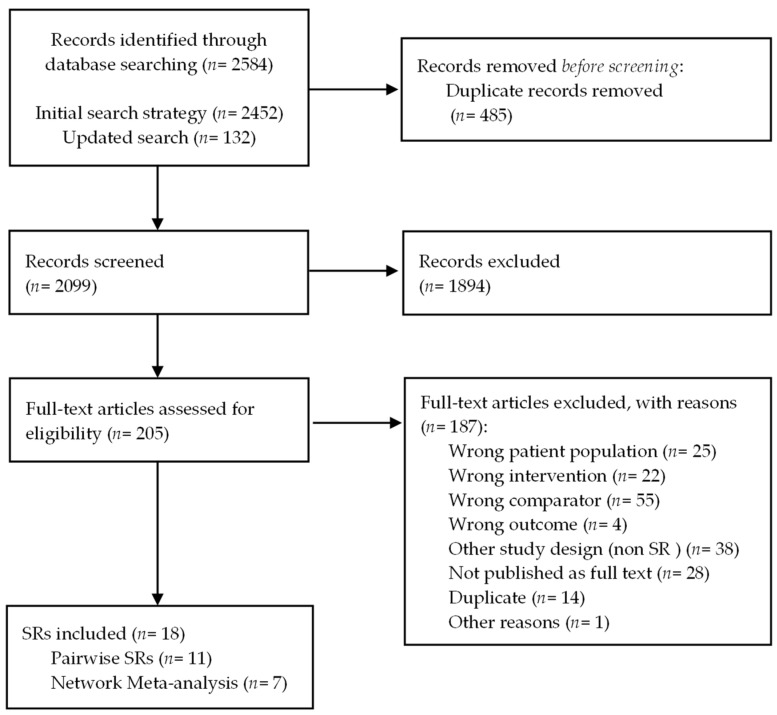
PRISMA flow diagram for the selection of studies.

**Figure 2 cancers-15-00766-f002:**
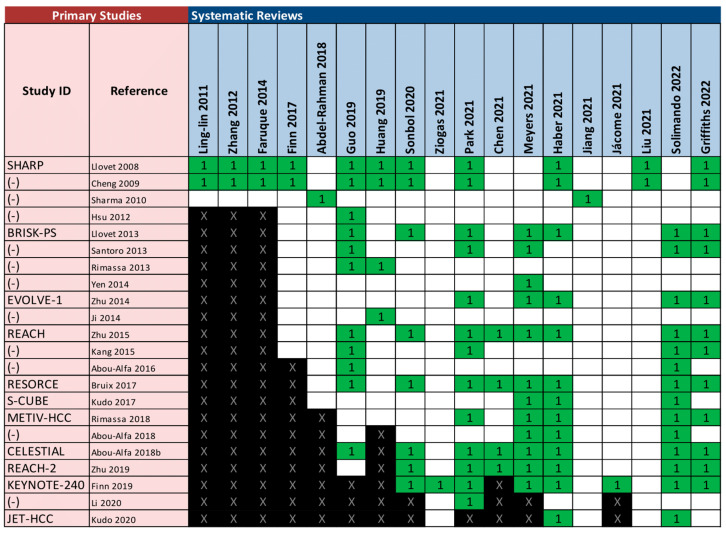
Matrix of evidence. All the included SRs are listed in the columns. All the relevant primary studies (i.e., those reporting at least one outcome of interest for the studied interventions) are listed in the rows. ‘1’ represents a primary study included in a specific SR. ‘X’ represents chronological structural missingness, that is, a primary study that cannot possibly be included in a SR, given that its publication date was more recent than the search date of the SR.

**Figure 3 cancers-15-00766-f003:**
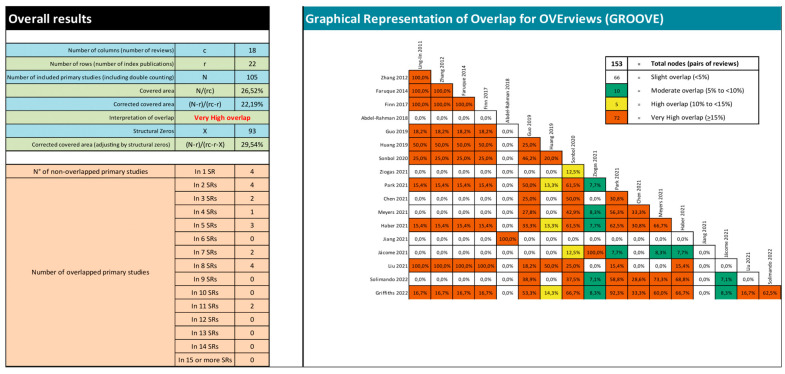
‘Graphical representation of overlap’ (GROOVE) assessment for each possible pair of SRs. For purposes of pairwise analysis of CCA, we did not consider structural missingness.

**Figure 4 cancers-15-00766-f004:**
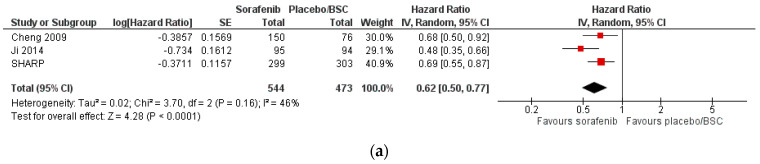
Forest plots for the outcomes ‘overall survival’ (**a**) and ‘toxicity’ (**b**) for the comparison of SOTs versus placebo/UPSC as first-line therapy in patients with advanced HCC.

**Figure 5 cancers-15-00766-f005:**
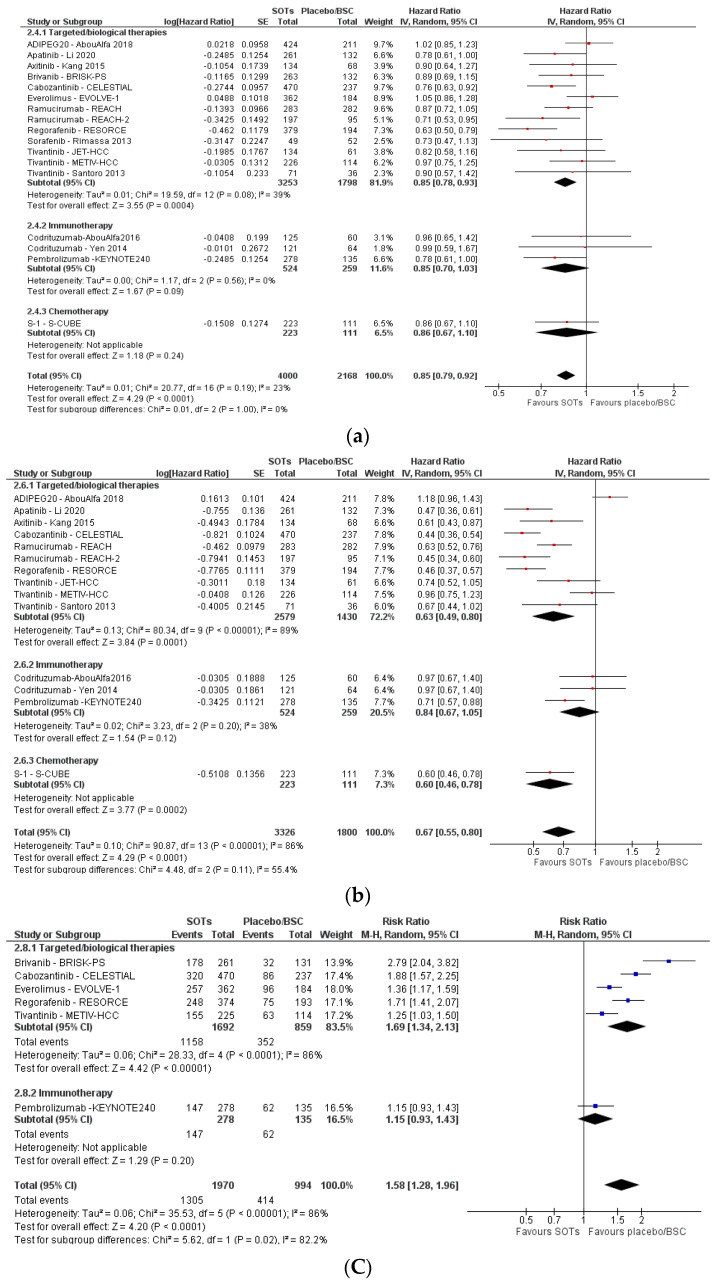
Forest plots for the outcomes ‘overall survival’ (**a**), ‘progression-free survival’ (**b**) and ‘toxicity’ (**c**) for the comparison of SOTs versus placebo/UPSC as second-line therapy or more in patients with advanced HCC.

**Figure 6 cancers-15-00766-f006:**
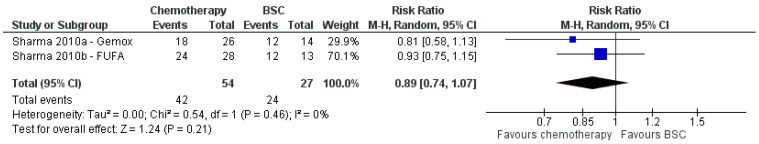
Forest plot for the outcome ‘overall survival’ for the comparison of chemotherapy versus placebo/UPSC in patients with advanced gallbladder cancer.

**Table 1 cancers-15-00766-t001:** Description of the included SRs.

Study ID	Country	Relevant Included Primary Studies ^1^/Total Included Studies	Type of Patients ^1^	Type of SOT	Line of Treatment	Comparator	Outcomes Assessed	Type of Meta-Analysis	Funding	CoI of the Review	Databases Searched	Search Timeframe
Ling-lin 2011 [53]	China	2/4	Advanced HCC	BIO/TT (Sorafenib)	First	Placebo	OS	Pairwise	NS	NS	MEDLINE, EMBASE, Cochrane, other	Until November 2019
Zhang 2012 [52]	China	2/6	Advanced HCC	BIO/TT (Sorafenib)	First	Placebo	OS, toxicity	Pairwise	Public	No CoI	MEDLINE, EMBASE, WoS	January 2005 to June 2011
Faruque 2014 [65]	Canada	2/72	Advanced HCC	BIO/TT (Sorafenib)	First	NS	OS, toxicity	Pairwise	Public and private	With CoI	MEDLINE, EMBASE, Cochrane	Until April 2012
Finn 2017 [49]	USA	2/14	Advanced HCC	BIO/TT (Sorafenib)	First	Placebo	OS	Pairwise	Private	With CoI	MEDLINE, EMBASE, CENTRAL, Scopus	Until February 2016
Abdel-Rahman 2018 [48]	Canada	1/7	Advanced gallbladder cancer	CT (Fluorouracil/folinic acid, or gemcitabine/oxaliplatin)	NS	BSC	OS, PFS, toxicity	Pairwise	No funding	No CoI	MEDLINE, EMBASE, CENTRAL, WoS, LILACS, clinicaltrials.gov	Until June 2017
Guo 2019 [51]	China	12/31	Advanced HCC	BIO/TT and IT (Sorafenib, vandetanib, brivanib, tivantinib, ramucirumab, axitinib, codrituzumab, cabozatinib)	First, second	Negative control (placebo or no treatment)	OS, PFS, toxicity	Network	NS	No CoI	MEDLINE, EMBASE, CENTRAL	Until March 2019
Huang 2019 [50]	China	5/11	Advanced HCC	BIO/TT (Sorafenib)	First, second	Placebo/BSC	OS, toxicity	Pairwise	NS	No CoI	MEDLINE, EMBASE, CENTRAL, WoS, clinicaltrials.gov	Until April 2018
Sonbol 2020 [64]	USA	8/14	Advanced HCC	BIO/TT and IT (Sorafenib, pembrolizumab, regorafenib, cabozantinib, ramucirumab, brivanib)	First, second	Placebo	OS, PFS	Network	NS	With CoI	MEDLINE, EMBASE, CENTRAL, WoS, Scopus	Until March 2020
Ziogas 2021 [63]	Greece	1/63	Advanced HCC	IT (Pembrolizumab)	Second	Placebo	OS, PFS, toxicity	Pairwise	NS	With CoI	MEDLINE, Cochrane, clinicaltrials.gov	Until November 2020
Park 2021 [62]	USA	13/24	Advanced HCC	BIO/TT and IT (Sorafenib, regorafenib, cabozitinib, ramucirumab, apatinib, pembrolizumab, brivanib, tivantinib, everolimus, axitinib)	First, second	Placebo	OS, PFS	Network	No funding	With CoI	MEDLINE, EMBASE, Cochrane	Until June 2020
Chen 2021 [59]	China	4/4	Advanced HCC	BIO/TT (Regorafenib, cabozantinib, ramucirumab)	Second	Placebo	OS, PFS, toxicity	Network	No funding	No CoI	MEDLINE, EMBASE, Cochrane	Until April 2019
Meyers 2021 [54]	Canada	12/49	Advanced HCC	BIO/TT and IT (Regorafenib, cabozantinib, brivanib, tivantinib, pembrolizumab, everolimus, ADI-peg 20, S-1, RO5137382/GC33)	Second or more	Placebo/BSC	OS, PFS, toxicity, QoL, symptoms	Pairwise	No funding	With CoI	MEDLINE, EMBASE, Cochrane	January 2000 to January 2020
Haber 2021 [61]	USA	13/49	Advanced HCC	BIO/TT and IT (Sorafenib, tivantinib, S-1, regorafenib, ramucirumab, ADI-PEG20, everolimus, cabozantinib, brivanib, pembrolizumab)	First, second	Placebo	OS, PFS	Pairwise	Public	With CoI	MEDLINE, Cochrane, WoS	June 2002 to December 2020
Jiang 2021 [58]	China	1/24	Advanced gallbladder cancer	CT (FUFA, or gemcitabine/oxaliplatin)	NS	BSC	OS, PFS, toxicity	Network	Public	NS	MEDLINE, EMBASE, Cochrane	Until August 2020
Jácome 2021 [57]	Brazil	1/3	Advanced HCC	IT (Pembrolizumab)	Second	Standard care	OS, PFS, toxicity	Pairwise	NS	With CoI	MEDLINE, CENTRAL, WoS, LILACS	Until February 2020
Liu 2021 [56]	China	2/15	Advanced HCC	BIO/TT (Sorafenib)	First	Placebo	OS, PFS, toxicity	Network	Public	No CoI	MEDLINE, EMBASE, CENTRAL, Cochrane, WoS, Scopus, other	Until August 2021
Solimando 2022 [60]	Italy	14/14	Advanced HCC	BIO/TT and IT (Tivantinib, S-1, regorafenib, ramucirumab, ADI-PEG20, everolimus, cabozantinib, brivanib, pembrolizumab, axitinib, codrituzumab)	Second	Placebo	OS, PFS	Network	Public	No CoI	MEDLINE, WoS, Scopus, clinicaltrials.gov	Until December 2020
Griffiths 2022 [55]	Canada	13/30	Advanced HCC	BIO/TT and IT (Sorafenib, tivantinib, regorafenib, ramucirumab, everolimus, cabozantinib, brivanib, pembrolizumab, axitinib)	First, second or more	Placebo	Toxicity	Pairwise	NS	With CoI	MEDLINE, EMBASE, CENTRAL	January 1990 to December 2021

^1^ We considered only relevant primary studies, defined as those that provided data for any comparison and outcome considered by this overview. BIO/TT: biological/targeted therapies; CoI: conflict of interest; CT: chemotherapy; IT: immunotherapy; NS: not specified; OS: overall survival; PFS: progression-free survival; QoL: quality of life; WoS: Web of Science 3.2. Methodological quality of the included reviews.

**Table 2 cancers-15-00766-t002:** Quality assessment of the included reviews according to AMSTAR-2.

Study ID	Q1	Q2 ^1^	Q3	Q4 ^1^	Q5	Q6	Q7 ^1^	Q8	Q9 ^1^	Q10	Q11 ^1^	Q12	Q13 ^1^	Q14	Q15 ^1^	Q16	Overall Quality of the Review
Ling-lin 2011 [53]	Y	N	Y	PY	Y	Y	N	PY	Y	N	Y	N	N	N	N	N	Critically low
Zhang 2012 [52]	Y	N	Y	PY	Y	N	N	Y	Y	N	Y	N	N	Y	Y	Y	Critically low
Faruque 2014 [65]	N	N	Y	PY	Y	Y	N	Y	Y	Y	Y	Y	Y	Y	Y	Y	Critically low
Finn 2017 [49]	Y	PY	N	Y	Y	N	N	Y	Y	N	Y	N	N	Y	Y	Y	Critically low
Abdel- Rahman 2018 [48]	Y	Y	Y	Y	Y	Y	Y	Y	Y	Y	Y	Y	Y	Y	Y	N	High
Guo 2019 [51]	Y	Y	Y	Y	Y	Y	N	Y	Y	N	NA	Y	Y	Y	Y	Y	Low
Huang 2019 [50]	Y	PY	N	Y	Y	Y	Y	Y	Y	N	Y	Y	Y	Y	Y	N	Moderate
Sonbol 2020 [64]	Y	N	N	PY	N	Y	N	PY	Y	N	Y	Y	Y	Y	N	Y	Critically low
Ziogas 2021 [63]	Y	N	N	Y	Y	Y	N	PY	N	N	NA	NA	N	Y	NA	Y	Critically low
Park 2021 [62]	Y	Y	Y	PY	PY	N	N	PY	Y	N	Y	Y	Y	N	Y	Y	Low
Chen 2021 [59]	Y	N	Y	PY	N	Y	N	PY	Y	N	Y	Y	Y	Y	Y	Y	Critically low
Meyers 2021 [54]	Y	N	Y	PY	N	N	N	PY	Y	N	NA	NA	N	N	NA	Y	Critically low
Haber 2021 [61]	Y	N	Y	PY	Y	Y	Y	PY	PY	N	NA	NA	Y	Y	NA	Y	Low
Jiang 2021 [58]	Y	N	Y	PY	N	Y	N	Y	Y	N	NA	NA	Y	Y	NA	Y	Critically low
Jácome 2021 [57]	Y	PY	Y	PY	Y	Y	N	N	Y	N	NA	NA	Y	Y	NA	N	Low
Liu 2021 [56]	Y	PY	Y	PY	Y	Y	N	PY	Y	N	Y	Y	Y	Y	Y	Y	Low
Solimando 2022 [60]	Y	N	Y	PY	N	Y	N	PY	Y	N	Y	Y	Y	Y	Y	N	Critically low
Griffiths 2022 [55]	Y	N	Y	Y	Y	Y	N	PY	Y	N	Y	Y	Y	N	N	Y	Critically low

^1^ Considered critical for AMSTAR-2 assessment; N: No; NA: Not applicable*; PY: Partial Yes; Y: Yes; Q1: Did the research questions and inclusion criteria for the review include the components of PICO? Q2: Did the report of the review contain an explicit statement that the review methods were established prior to the conduct of the review and did the report justify any significant deviations from the protocol? Q3: Did the review authors explain their selection of the study designs for inclusion in the review? Q4: Did the review authors use a comprehensive literature search strategy? Q5: Did the review authors perform study selection in duplicate? Q6: Did the review authors perform data extraction in duplicate? Q7: Did the review authors provide a list of excluded studies and justify the exclusions? Q8: Did the review authors describe the included studies in adequate detail? Q9: Did the review authors use a satisfactory technique for assessing the risk of bias (RoB) in individual studies that were included in the review? Q10: Did the review authors report on the sources of funding for the studies included in the review? Q11: If meta-analysis was performed did the review authors use appropriate methods for statistical combination of results? Q12: If meta-analysis was performed, did the review authors assess the potential impact of RoB in individual studies on the results of the meta-analysis or other evidence synthesis? Q13: Did the review authors account for RoB in individual studies when interpreting/ discussing the results of the review? Q14: Did the review authors provide a satisfactory explanation for, and discussion of, any heterogeneity observed in the results of the review? Q15: If they performed quantitative synthesis did the review authors carry out an adequate investigation of publication bias (small study bias) and discuss its likely impact on the results of the review? Q16: Did the review authors report any potential sources of conflict of interest, including any funding they received for conducting the review? * We assessed some of the answers related to meta-analysis as NA if the review had no meta-analysis, had only one included study relevant for our question or if the conducted meta-analyses were not related to the scope of this overview.

## Data Availability

The supplementary data presented in this study are openly available in Open Science Framework at https://doi.org/10.17605/OSF.IO/7CHX6 (accessed on 9 November 2022). Other data are available through contact with the corresponding author at reasonable request.

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
