# Peer review of "Systemic Oncological Treatments versus Supportive Care for Patients with Advanced Hepatobiliary Cancers: An Overview of Systematic Reviews"

_cancers, 2023, doi:10.3390/cancers15030766_

Round 1
Reviewer 1 Report
Relevant, systematized study with practical implications.
They refer that "We included SRs assessing the clinical impact of SOTs in advanced HBCs, published from 2008 onwards (since one of the first landmark studies for our question was published this year)." - Line 107
On line 150 authors say - "We did not use any language or publication status restrictions." - Here, it should be clear whether the time limit was taken into account.
Methodologically, the study seems quite robust to me; it makes a detailed description of the method used, the selection and analysis process, and what criteria were defined.
Given the volume of information, which is excellent for the richness of detail, I suggest introducing bullet points to show the reader the review's main findings.
It was possible to observe the research strategy, which seems to be quite comprehensive but specific.
Reviewer 2 Report
This is an overview of systematic reviews to assess the effectiveness of systemic oncological treatments versus best supportive in patients with primary advanced hepatobiliary cancers. Methodology showed substantial rigor consistent with Cochrane overviews. Specifically, the search terms were appropriate and thorough showing sensitivity and specificity for capturing the relevant literature. Five databases were searched consistent with recommendations for performing systematic reviews. Overlap of primary studies between systematic reviews was monitored. Meta-analysis was performed and risk of bias was assessed, quality of evidence appropriately applied the AMSTAR tool the certainty of evidence was assessed using the GRADE checklist. Only 18 systematic reviews met final inclusion criteria, but this exceeds the minimum for a meta-analysis and appears to be a valid number considering the specific research questions and population of interest. The findings highlight the need for further research of higher quality for these cancers, especially as it relates to toxicity from therapies and importance of supportive care. Overall, I found this overview to be of reliable quality and feel the topic and the gap noted to be important to clinicians and researchers alike who work with patients with hepatobiliary cancers and appropriate for publication in Cancers. It can also serve as a guidepost for overviews of systematic reviews. I recommend acceptance for publication.
Reviewer 3 Report
This systematic review (SR) of SRs aims at synthesizing the effects of several outcomes, including quality of life and functional status, in addition to survival for patients with advanced hepatobiliary cancers (HBCs). The following points will help strengthen the manuscript.
1. Lines 107-113: Please be clear if SRs that only included randomized controlled trials were eligible for this SR of SRs. If not, please specify which of the experimental and non-experimental study designs were part of the eligibility criteria?
2. Line 118-121: Is it “primary and recurrent” HBCs as in here or only “primary” (line 24) as in the Abstract?
3. Lines 130-136: Please be explicit here if it is the SR or “studies” that some SRs included are being referenced here in terms of the comparator group descriptions.
4. Lines 149-150: Which database in “Cochrane Library”? Please be specific.
5. Lines 152-156: Please consider including search strategies for all databases given the importance of transparency in a SR of SRs.
6. Lines 157-162: Please clearly state the 2 screening author initials (now there are a total of 11 initials) and 1 author initial who resolved disagreements (now there are a total of 4 initials). Please do the same for full-text screening.
7. Lines 164-169: Again, please be clear about author initials for data extraction. It is concerning to this reviewer that data extraction was not completed by 2 authors independently.
8. Lines 170-175: Same as above. Please be clear about author initials. It is concerning to this reviewer that risk of bias was not completed by 2 authors independently.
9. Lines 173-174: Which “Cochrane RoB tools” were prioritized?
10. Lines 194-196: Could authors provide a reference?
11. Lines 200-202: :Funnel plots”?
12. Lines 203-208: “SoF” table? Please ensure transparency. It is written out later in line 312 but that is not very helpful for this section.
13. Lines 214-215 & Figure 1: First, please note that this is not the updated flow diagram. Second, please explain what do “studies” refer to in this diagram. Third, please explain what is meant by “study design”? “Publication type”?
14. Line 221- 223: It is the first time “network meta-analysis” is mentioned. Thanks for including information afterwards but please consider revising earlier relevant sections such as revising PRISMA flow diagram further to incorporate this information.
15. Lines 224-231 & Table 1: Please consider revising Table 1 to (a) ensure that it is easier to distinguish between entries, (b) incorporate both primary studies that were included in the respective SRs and those that are included in this SR of SRs, adding primary study design information, (c) include reference number for entries and (d) exclude information such as “grey literature” which does not seem to be link to any of the discussions afterwards.
16. Lines 285: “Trials”?
17. Lines 334-341 & 367-371: Authors make references to results from a number of SRs (with overall “low” or “critically low” assessments in Table 2, one SR even gets a “not applicable” mark in terms of its meta-analysis?) in their Results section without providing any details of the network analyses that these SRs conducted. Please provide a discussion about the need to incorporate their results in this manuscript’s Results section and, at least, provide some guidance as to how valid are the network analyses and how reliable are their results if that information will be kept in there.
18. Please avoid typos and ensure completeness and transparency in the manuscript: a) please write out all acronyms (such as “ECOG”, line 140 or “ESMO”, “ASCO” in line 411) the first time they are mentioned, b) please ensure that all tables and figures are self contained (see, for example, Figure 1, which PRISMA flow diagram is this, please provide a reference) and c) “Web of Scienc”?
Reviewer 4 Report
The current reviewer has several concerns on the validity of the analysis that appears indeed as a "melting pot" of different treatment and diseases. I would suggest to focus the paper only on HCC (only two included studies were on GB cancer). Anyway, i don't see any point in merging different drugs with different mechanism of action.
Reviewer 5 Report
-A meticulous well performed study of a relative rare disease
-I agree completely with the warning of the authors that OS and PFS are not "the only endpoints" we need to evaluate in this very unfavorable diseases, much more important is the QoL /side effects / performance status ...in this short survival time...
Round 2
Reviewer 3 Report
Thanks for the revised manuscript.
The following points need further attention. Old numbering is kept for convenience.
4. Old “Lines 149-150: Which database in “Cochrane Library”? Please be specific.”
Thanks but this reviewer is not sure why CENTRAL was searched; no rationale is given.
6. Old “Lines 157-162: Please clearly state the 2 screening author initials (now there are a total of 11 initials) and 1 author initial who resolved disagreements (now there are a total of 4 initials). Please do the same for full-text screening.”
This reviewer understands the scope of the larger project. However, this reviewer is still concerned that any of the 2 authors among a total of 11 initials included have screened and any of the 1 author among a total 4 initials included have resolved disagreements for this SR of SRs. And this is only the screening phase. Could authors add a separate Strengths and Limitations section (which is now embedded in the Discussion section) to explicitly discuss how much of a heterogeneity/bias this (that is, the fact that there was no consistency in terms of the authors who conducted the screening phase) could have introduced to this SR of SRs so that the readership is introduced to an explicit discussion about this limitation? Similarly, please also add a separate discussion for the processes used for the full-text screening phase and the limitations that might have resulted.
7. Old “Lines 164-169: Again, please be clear about author initials for data extraction. It is concerning to this reviewer that data extraction was not completed by 2 authors independently.”
Again, as above, please also include a discussion in the Strengths and Limitations section about the heterogeneity/bias that could be introduced because of the processes followed for data extraction phase (this time also add the fact that the data extraction phase was not completed by 2 authors independently).
8. Old “Lines 170-175: Same as above. Please be clear about author initials. It is concerning to this reviewer that risk of bias was not completed by 2 authors independently.”
Again, please see above. The same concerns persist for the risk of bias phase.
11. Old “Lines 200-202: :Funnel plots”?”
Thanks for indicating that this is “corrected” in the internal response. Should this reviewer now assume that funnel plots were added to the supplementary material?
13. Old “Lines 214-215 & Figure 1: First, please note that this is not the updated flow diagram. Second, please explain what do “studies” refer to in this diagram. Third, please explain what is meant by “study design”? “Publication type”?”
Thanks for highlighting the first paragraph of the Results section. However, first, Figure 1 needs to be updated to include the current flow diagram (also see #18 below in terms of providing a reference). Second, Figure 1 needs to explicitly indicate how many “reviews” were included in this SR of SRs. Third, thanks for adding to the methods section but please be transparent in Figure 1 itself in terms of what is meant by “study design” & “publication type”.
14. Old “Line 221- 223: It is the first time “network meta-analysis” is mentioned. Thanks for including information afterwards but please consider revising earlier relevant sections such as revising PRISMA flow diagram further to incorporate this information.”
Thanks for stating this in the Methods section now. Please add your rationale for incorporating SRs with “network meta-analysis” in this SR of SRs upfront and clearly explain how results from these SRs were used. Also, please consider incorporating this information to Figure 1.
16. Old “Lines 285: “Trials”?”
Thanks, now it is clear that only RCTs were included in the SRs considered.
17. Old “Lines 334-341 & 367-371: Authors make references to results from a number of SRs (with overall “low” or “critically low” assessments in Table 2, one SR is even noted as “not applicable” in terms of its meta-analysis?) in their Results section without providing any details of the network analyses that these SRs conducted. Please provide a discussion about the need to incorporate their results in this manuscript’s Results section and, at least, provide some guidance as to how reliable are these results if that information will be kept in there.”
Thanks for internal response. This reviewer still finds it confusing how the results from the network analysis of the SR which is noted as “not applicable” in terms of its meta-analysis in Table 2 is discussed in the Results section.
18. Old “Please avoid typos and ensure completeness and transparency in the manuscript: a) please write out all acronyms (such as “ECOG”, line 140 or “ESMO”, “ASCO” in line 411) the first time they are mentioned, b) please ensure that all tables and figures are self contained (see, for example, Figure 1, which PRISMA flow diagram is this, please provide a reference) and c) “Web of Scienc”?”
Thanks; please see above review point #13 for b).
Reviewer 4 Report
The authors did not reply satisfactorily to my comment. The serious issue of the heterogeneity still remains.
